# Epidemiological and Clinical Manifestations of Acute Rheumatic Fever in Far North Queensland, Australia

**DOI:** 10.3390/pathogens14050442

**Published:** 2025-04-30

**Authors:** Mia Crous, Allison Hempenstall, Nancy Lui-Gamia, Caroline Taunton, Josh Hanson

**Affiliations:** 1Cairns and Hinterland Hospital and Health Service, Cairns, QLD 4870, Australia; mia.crous@health.qld.gov.au; 2Torres and Cape Public Health Unit, Cairns, QLD 4870, Australia; allison.hempenstall@health.qld.gov.au (A.H.); nancy.lui-gamia@health.qld.gov.au (N.L.-G.); caroline.taunton@health.qld.gov.au (C.T.); 3Kirby Institute, University of New South Wales, Kensington, NSW 2052, Australia

**Keywords:** acute rheumatic fever, rheumatic heart disease, First Nations Health, epidemiology

## Abstract

We used the Queensland acute rheumatic fever (ARF) and rheumatic heart disease (RHD) register to describe the clinical phenotype and the characteristics of individuals diagnosed with ARF in Far North Queensland, Australia, between January 2012 and December 2023. There were 830 episodes of ARF in 740 individuals during the study period; 785/830 (95%) episodes occurred in First Nations Australians and 696/824 occurred in areas of socioeconomic disadvantage. There was no significant change in the overall incidence of ARF during the study period (Spearman’s rho = 0.51, *p* = 0.09). The median (interquartile range) age of the cohort was 15 (10–23) years, although 276/830 (33%) episodes of ARF occurred in individuals ≥ 20 years. Individuals with carditis, polyarthritis, an abnormal electrocardiogram, fever and elevated inflammatory markers were more likely to have confirmed ARF. The presence of polyarthralgia, monoarthritis or skin manifestations was not associated with a diagnosis of confirmed ARF. Individuals with monoarthralgia were less likely to have confirmed ARF. At the end of the study period, 264/706 (37%) individuals who had access to echocardiography had confirmed RHD. Individuals who did not have echocardiography documented as a component of their initial episode of care were more likely to have severe RHD at the end of the study (25/339 (7%) versus 7/401 (2%), *p* < 0.0001). ARF and RHD continue to be diagnosed in First Nations Australians in tropical Australia. It seems unlikely that Australia will achieve its stated aim of eliminating RHD by 2031.

## 1. Introduction

Acute Rheumatic Fever (ARF) is an immune-mediated inflammatory condition that can follow Group A *Streptococcus pyogenes* (GAS) infection [1]. ARF can involve the skin, subcutaneous tissue, joints, brain, and the heart [2]. Severe or recurrent episodes of ARF-induced carditis can result in permanent cardiac damage, a condition termed rheumatic heart disease (RHD). RHD is common and can be life-threatening; it was estimated that globally in 2019 there were 40.5 million people living with RHD and that it caused almost 310,000 deaths [3]. Improvements in the standard of living and earlier treatment of GAS infections mean that ARF and RHD are now rarely diagnosed in resource-rich settings [4].

However, Australia is a notable exception to this trend [5]. The burden of ARF and RHD in Australia is borne almost entirely by its Aboriginal and Torres Strait Islander peoples (hereafter respectfully referred to as First Nations peoples). While the overall incidence of ARF in Australia is 69 per 100,000, the incidence in First Nations peoples is up to 121 times that of other Australians [6]. Despite Australia’s well-resourced, universal health system, the incidence of ARF in Australia’s First Nations peoples continues to rise. This, in turn, increases the likelihood of further rises in the prevalence of RHD, which in some First Nations communities may exceed 5% [7,8].

The RHD Endgame Strategy aims to eliminate RHD in Australia by 2031 by ensuring the optimal delivery of primary, secondary and tertiary prevention strategies [5]. A key element of this strategy is the prompt, accurate diagnosis of ARF and the subsequent administration of secondary antibiotic prophylaxis to prevent GAS infection and RHD progression [5,9]. However, up to 78% of Australians diagnosed with RHD have no prior diagnosis of ARF suggesting that many episodes of ARF are currently missed [10]. This is likely to be explained, at least in part, by a failure to recognise ARF presentations when they occur.

However, the diagnosis of ARF—which can include a variety of dermatological, rheumatological, neurological and cardiac manifestations—can be challenging [11]. While the revised Jones criteria recognise the diverse clinical presentations of ARF, they have imperfect sensitivity and specificity and their clinical utility is influenced strongly by the pre-test probability of ARF in the examined population [12,13,14,15]. It is therefore essential for clinicians to understand the characteristic presentations of ARF in their local clinical setting to enable the prompt delivery of the suite of interventions that reduce the risk of a future RHD diagnosis [15].

The region of Far North Queensland (FNQ) in tropical northeast Australia has a significant—and increasing—incidence of RHD; nearly all of its cases are seen in its First Nations peoples [16]. ARF and RHD are both notifiable conditions in Queensland, but almost half of the individuals diagnosed with RHD in the FNQ region have no recorded history of ARF, suggesting that local clinicians may be missing opportunities to recognise and diagnose the condition [16]. This study examined routinely collected epidemiological and clinical data from the Queensland ARF and RHD register to characterise ARF presentations in the FNQ population. Once disseminated, these data would be expected to facilitate the prompt local diagnosis of ARF and the more timely delivery of the multi-faceted care that is required to reduce the local incidence of RHD and its complications [15,16,17].

## 2. Methods

### 2.1. Setting

FNQ is a region of 380,000 km^2^ with a population of approximately 290,000 people. The majority of FNQ residents are Caucasian, however, about 49,000 (17%) identify as First Nations people [18]. Indeed, the FNQ region is the only region of Australia that has the homelands of both Aboriginal and Torres Strait Islander Australians. FNQ shares a maritime border with Papua New Guinea and the FNQ region has a larger number of individuals born in Papua New Guinea than any other region in Australia. Recent migration from Asia has also seen an increase in the number of FNQ residents who were born in South and Southeast Asia [18].

The FNQ region is served by two distinct public health services: the Cairns and Hinterland Hospital and Health Service (CHHHS) which serves approximately 265,000 people (of whom about 32,000 (12%) identify as First Nations people) living in the region surrounding the administrative hub of Cairns, and the Torres and Cape Hospital and Health Service (TCHHS) which serves a rural and remote population of approximately 25,000 people (of whom approximately 17,000 (69%) identify as First Nations people) living on the Cape York Peninsula or the Torres Strait Islands (Figure 1).

### 2.2. Data Collection

ARF and RHD have been notifiable conditions in Queensland since 1999 and 2018, respectively; the demographic and clinical data of notified cases are recorded on the Queensland ARF and RHD register. In this study, the register was interrogated to identify FNQ residents with a diagnosis of confirmed, probable or possible ARF diagnosis between 1 January 2012 and 31 December 2023. Episodes of ARF > 6 months apart in the same individual were included as separate presentations. This time period was chosen as the register was missing some crucial data before 2012.

The demographic characteristics of the individual presenting with each episode of ARF were recorded. Socioeconomic disadvantage was defined using the Socio-Economic Indexes for Area (SEIFA) Index of Relative Socio-Economic Advantage and Disadvantage (ISRAD) score, the Australian Bureau of Statistics’ measure of socioeconomic disadvantage [20]. The revised Jones criteria that were satisfied in each case of ARF were recorded and used to define whether the case was confirmed, probable or possible (Table 1). If a criterion was not recorded, it was presumed to be absent. High- and low-risk populations were defined using the 2020 Australian guidelines for the prevention, diagnosis and management of acute rheumatic fever [15]. The register was also interrogated to determine the presence and severity of RHD in each individual at the end of the study period. The presence and severity of RHD were defined using the 2023 World Heart Federation guidelines [21].

### 2.3. Statistical Analysis

De-identified data were extracted and analysed using statistical software (Stata version 18). Groups were compared using logistic regression and trends over time were determined using Spearman’s test for correlation. The distributions of age and the titres of serological tests were non-parametric and so they are presented as the median (interquartile range (IQR)). If individuals were missing data, they were not included in analyses that evaluated those variables. Australian Bureau of Statistics data were used to calculate disease incidence [18].

## 3. Results

There were 830 notifications of confirmed, probable and possible ARF to the register during the study period; 492 (56%) were confirmed, 180 (20%) were probable and 158 (18%) were possible. The 830 episodes of ARF occurred in 740 individuals. There was enough information in the register to determine whether an episode of ARF was an initial or a recurrent episode in 777/830 (94%) presentations; 191/777 (25%) were recurrent episodes.

The overall annual incidence of ARF in the FNQ population in 2023 (the end of the study period) was 29.1/100,000 population. The incidence of confirmed ARF in FNQ was 9.7/100,000 population. There was no significant change in the overall incidence of ARF during the 2012–2023 study period (r_s_ = 0.51, *p* = 0.09), but the proportion of episodes that satisfied the criteria for confirmed ARF declined (r_s_ = −0.31, *p* < 0.0001) (Figure 2).

At the end of the study period, the annual overall incidence of ARF in the FNQ First Nations Australian population was 157.9/100,000 population and the annual overall incidence in the TCHHS population was 200.6/100,000 population. Meanwhile, the annual incidence of confirmed ARF in the FNQ First Nations Australian population was 59.2/100,000 population and the annual incidence of confirmed ARF in the TCHHS population was 83.6/100,000 population.

Of the 830 episodes, 459 (55%) occurred in the TCHHS and 371 (45%) in the CHHHS (Table 2, Figure 3). The median (interquartile range, range) age of the patient at the time of their ARF episode was 15 (10–23, 3–55) years; a diagnosis of confirmed ARF was more common in individuals aged 5–14 (odds ratio (OR), (95% confidence interval (95% CI)): 2.64 (1.98–3.53), *p* < 0.001 (Figure 4).

Of the 830 episodes of ARF, 785 (95%) occurred in First Nations people; First Nations people were more likely to present with recurrent ARF than non-First Nations people (188/734 (26%) versus 3/43 (7%), OR (95% CI): 4.59 (1.40–15.01), *p* = 0.01). Of the 830 presentations, 512 (62%) occurred in individuals residing in regions with a SEIFA score in the lowest decile (Figure 5). Individuals residing in regions with a SEIFA score in the lowest decile were no more likely to present with recurrent ARF than individuals living in other locations (116/478 (24%) versus 75/299 (25%), OR (95% CI): 0.96 (0.68–1.34), *p* = 0.80). The clinical and laboratory characteristics of the cases are presented in Table 3.

Only 441/830 (53%) episodes of ARF had an echocardiogram associated with the presentation documented in the register and in 374/441 (85%) there was no evidence of RHD. However, by the end of the study period, 264/740 (36%) individuals with an episode of ARF during the study period had confirmed RHD (mild, moderate or severe disease); this included 32/740 (4%) with severe RHD (Figure 6). Individuals who did not have a documented echocardiogram associated with the ARF episode were more likely to have severe RHD at the end of the study period (25/339 (7%) versus 7/401 (2%), (OR (95% CI): 4.48 (1.91–10.50), *p* < 0.0001). Individuals who had presented with recurrent ARF were more likely to have RHD (OR (95% CI): 2.45 (1.61–3.72), *p* < 0.0001) and more likely to have severe RHD at the end of the study period than those who did not have recurrent ARF (OR (95% CI): 5.55 (2.53–12.18), *p* < 0.0001). At the end of the study period, the median (IQR) age of the 32 individuals with severe RHD was 27 (17–36) years; 21/32 (66%) were female.

## 4. Discussion

This study examined the characteristics of individuals presenting with ARF in the FNQ region of tropical Australia and highlights the clinical and laboratory findings that are most associated with a diagnosis of confirmed ARF. It emphasises the importance of prompt echocardiography in patients presenting with symptoms of ARF. Individuals in the cohort with carditis were almost five times as likely to have a diagnosis of confirmed ARF than those without this finding. And individuals who did not have echocardiography documented were more likely to develop severe RHD. The study also found that joint symptoms that represented a major Jones criterion were present in over 80% of the ARF episodes, but only polyarthritis—and not monoarthritis or polyarthralgia—was associated with a diagnosis of confirmed ARF. Finally, fever, an elevation in inflammatory markers and higher titres of antibodies against GAS also increased—while monoarthralgia decreased—the likelihood of a confirmed diagnosis. These findings provide data that will facilitate the prompt local diagnosis of in the region and the delivery of the multi-faceted care that is required to reduce the local incidence of RHD and its complications.

The international literature suggests that carditis is present in 50–70% of individuals during their first episode of ARF [12,22]. Although carditis was diagnosed in less than 13% of the cohort it was the manifestation that was most associated with a diagnosis of confirmed ARF. The lower proportion of individuals with a diagnosis of carditis in our cohort may be at least partly explained by the challenges with providing timely echocardiography across the region’s 380,000 km^2^ expanse. Just over half of the ARF episodes had echocardiography documented as part of their assessment and the patients who did not have echocardiography were more likely to have severe RHD at the end of the study period. Earlier recognition of cardiac involvement in these individuals may have resulted in intensified efforts to deliver secondary prophylaxis to reduce disease progression [9,23,24]. The fact that there was no increase in the number of individuals having echocardiography performed as part of their assessment during the study period would appear to be a useful focus for the local RHD programme as it expands [16].

Our study highlights the importance of considering ARF as a diagnosis in individuals presenting with joint symptoms, particularly in those at higher risk of the disease [12,15]. The proportion of individuals in the cohort with joint symptoms is similar to other Australian and international series [12,25,26,27], although it was notable that individuals with polyarthritis were more likely to have a confirmed ARF diagnosis, which was not the case in individuals with polyarthralgia or monoarthritis. Individuals with monoarthralgia were less likely to have a confirmed ARF diagnosis, but this is likely to be at least partly explained by increased awareness of ARF among local clinicians and a lower threshold for notification [16,23]. The suggestion is supported by the fact that the number of cases of confirmed ARF declined significantly over the study period.

Erythema marginatum and subcutaneous nodules were documented in less than 4% of the cases, which is, again, similar to the rates seen in the international literature, although it may also represent a lack of familiarity with these manifestations of ARF and challenges in their recognition in a population with more pigmented skin and a higher burden of other skin conditions [12,28,29]. Although erythema marginatum and subcutaneous nodules have been described as highly specific manifestations of ARF, there was no association in this cohort between the presence of these manifestations and a confirmed diagnosis of ARF.

Sydenham’s chorea is documented in approximately 10–30% of episodes of ARF in the international literature and in 19% and 31% of cases of ARF in other Australian jurisdictions [12,22,27], Sydenham’s chorea was seen in only 6% of this cohort (although it was seen in 11% of those with confirmed disease) The lower rate of chorea in our cohort compared with other Australian cohorts may represent local clinicians’ lack of familiarity with this unique manifestation of ARF, over-reporting of conditions that were not ARF (generating a greater denominator) or genetic differences in the population of the FNQ region (over 40% of our cohort had Torres Strait Islander heritage) [30].

Most series focus on the prevalence of the different major manifestations of ARF, which is understandable, however, the minor criteria in our cohort were also helpful in establishing a diagnosis of confirmed ARF. Fever, inflammatory markers and a prolonged PR interval were all more common in patients with confirmed ARF, while monoarthralgia was less common. It was also notable that individuals with confirmed ARF were also more likely to have a higher titre of ASOT and anti-DNase B antibodies. An elevated CRP or ESR was approximately twice as common in individuals with confirmed ARF than those with possible ARF and the median ASOT in individuals with confirmed ARF was almost twice as high. Laboratory data are more easily accessible in remote clinics in the FNQ region and may help local clinicians identify the higher-risk patients—most in need of enhanced follow-up—while echocardiography is awaited.

Our study also provides data to support the adds to the growing body of evidence from Australia and New Zealand that suggest a role for GAS infection of the skin in the pathogenesis of ARF [31,32]. Over 5% of individuals had isolation of GAS from a skin lesion before, or at, the time that they presented with ARF symptoms. Meanwhile, a proportion of the 20% with a stated history of skin sores or sore throat would have only had skin sores. A temporal association between the isolation of GAS from a skin lesion and a diagnosis of ARF does not necessarily imply a pathophysiological role of skin infection, however, there are compelling epidemiological data to suggest that GAS skin infection may make an important contribution to the pathogenesis of ARF, particularly in tropical settings [33]. Given the many other benefits of improved skin health, there are a few downsides to including efforts to address skin health in public health strategies to reduce ARF incidence [15,34].

The greatest number of ARF episodes occurred, as expected, in children aged 5–14 years, although it was notable that more than 43% of all cases of confirmed ARF in the cohort occurred in individuals outside of this age group. It is important to consider ARF in older individuals in the appropriate clinical context to ensure that they receive prompt investigation and treatment, although it remains important to actively exclude differential diagnoses, including connective tissue disorders and infection [35,36,37,38]. There is a significant local incidence of rheumatological disorders and septic arthritis in the local FNQ population and local First Nations people are also disproportionately affected by these conditions [39,40,41]. Septic arthritis, in particular, should always be considered particularly in those presenting with monoarthritis and may need empirical antibiotic cover until the diagnosis is excluded [15].

The overall incidence of ARF did not decline during the study period which aligns with other recent Australian data that highlight a stable, or increasing, incidence of ARF/RHD. This is despite the implementation of a national strategy to address and eliminate these preventable diseases [7,42]. It is likely that this finding is explained, at least partly, by improved case detection and notification and the significant decline in confirmed episodes of ARF in this cohort during the study period provides evidence for this [16,23]. However, our data highlight the critical need for interventions that target primordial and primary prevention of ARF.

In recent years Australian programmes have focussed on primary, secondary and tertiary prevention strategies to reduce the incidence of ARF and RHD [15]. These programmes have resulted in improved adherence to secondary prophylaxis which has made some impact on reducing progression to severe RHD across Australia [23]. Improved adherence to secondary prophylaxis would also be anticipated to translate into a decrease in recurrent ARF [43]. However, there is a growing recognition that without similar efforts to address the social determinants of health that drive the incidence of ARF, there is likely to be limited progress [23,44,45].

ARF is the archetypal disease of disadvantage, and our study provides further data, as if this were needed, to highlight its contribution to the incidence of ARF and RHD in 21st century Australia. FNQ has three of the most disadvantaged Local Government Areas in Australia and this disadvantage, borne disproportionately by First Nations people living in the region, has an important impact on the local incidence of infectious diseases [16,46,47,48]. Over 60% of the cohort lived in a region in the lowest decile of the SEIFA Index of Relative Socio-Economic Advantage and Disadvantage score while almost 95% of the cohort identified as First Nations people (despite First Nations people comprising only 17% of the local population). Over 55% of the episodes occurred in the TCHHS which serves the remote Cape York Peninsula and Torres Straits islands, although less than 10% of the FNQ population lives in the region. These data again highlight the primacy of the social determinants of health in the incidence of ARF, RHD and its clinical course [15,16,44,49].

The study had several important limitations. Although the primary source for the data was the Queensland ARF and RHD register, the imperfect specificity of the revised Jones criteria means that this cohort almost certainly contains individuals who did not have ARF. Furthermore, although ARF has been a notifiable disease in Queensland since 1998, many cases are not recognised or are not reported to the register as evidenced by the fact that locally, approximately 50% of the individuals diagnosed with RHD have no recorded history of ARF [16,50]. While the register is maintained diligently, its data quality depends on the thoroughness of the clinical assessment and, even more importantly, on its documentation. The retrospective nature of this study precluded comprehensive data collection, and we were unable to determine how actively clinical data had been sought. It was not always clear whether, for any given criterion, there was no abnormality or no data available. The clinical assessment will necessarily depend on the clinical experience of the attending health worker and their familiarity with ARF. Certainly, anecdotally, a detailed skin assessment is not performed in all FNQ patients presenting with ARF symptoms and electrocardiograms may not always be performed at the time of presentation. Knowing that manifestations of ARF were absent despite being specifically sought by an experienced clinician would have helped characterised the clinical phenotype of ARF in the region more reliably. Our study also did not examine the impact of treatment of symptoms on the clinical phenotype. Although monoarthritis was not associated with a diagnosis of confirmed ARF in this cohort, patients treated with non-steroidal anti-inflammatory drugs early in the course of ARF, particularly before the other signs and symptoms of ARF become distinct, may not develop polyarthritis [14]. In patients presenting with recurrent ARF, we did not examine factors associated with their adherence to secondary prophylaxis which would, of course, be expected to reduce the risk of recurrent ARF. We did not stratify the presentation of our findings by initial and recurrent presentation or whether the individual came from a high- or low-risk population, although over 98% of individuals in the cohort would be considered at high risk of the disease [15].

Acknowledging these limitations, the study provides contemporary insights into the characteristics of individuals with ARF in this region of tropical Australia, their clinical findings at presentation and it identifies which of these clinical findings were most associated with a diagnosis of confirmed ARF. The study also demonstrates that despite greater awareness and increased financial investment the overall incidence of ARF in the FNQ region is not declining, which will necessarily mean new cases of RHD with its associated morbidity and premature mortality. Over a third of the cohort had established RHD at the end of the study period and over 4% had severe disease, a dispiriting finding in 21st century Australia.

It is hoped that this study has provided data that will improve the recognition and treatment of ARF in the region, reducing the risk of progression to RHD. Future prospective studies might examine individual and systemic factors that delay individuals’ presentation, which could, in turn, inform strategies to expedite their presentation for diagnosis and management. Evaluation of local individual and systemic factors that affect adherence to secondary prophylaxis would also help develop strategies to reduce the risk of recurrent ARF, which represented almost 25% of the ARF presentations in this series.

## 5. Conclusions

This study provides contemporary data on the clinical presentations of individuals with ARF in the FNQ region of Australia and the clinical and laboratory findings that are most associated with a diagnosis of confirmed ARF. The study emphasises the importance of prompt electrocardiography and echocardiography, provides insights into the common patterns of joint involvement, and demonstrates the value of laboratory data in helping secure the diagnosis of ARF. The study also highlights that while local children aged 5–14 are the most likely to have ARF, confirmed ARF is not uncommon in older populations. It also adds to the growing body of data to suggest that there is a role for GAS infection of the skin in the pathogenesis of ARF [33].

However, the study also emphasises that the most important risk factor for ARF is the socioeconomic disadvantage that many First Nations people in FNQ continue to experience. It is therefore essential that policy makers developing public health strategies to reduce the local burden of RHD work with local communities to also address the social determinants of health that are the underlying cause of ARF—and many other communicable and non-communicable diseases—in the region. Until we address the underlying inequity that drives the incidence of ARF and RHD in Australia, progress will be limited. We need to do better if we are to reach our collective goal of eliminating RHD in Australia by 2031.

## Figures and Tables

**Figure 1 pathogens-14-00442-f001:**
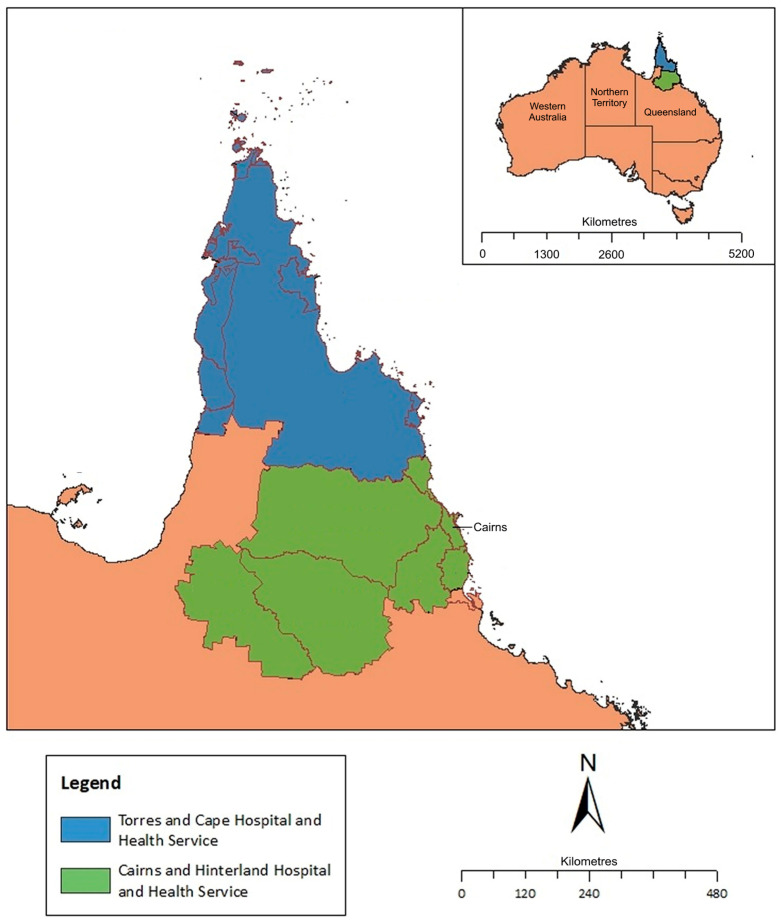
Map of Far North Queensland, Australia, showing catchment area for current study. Image adapted from Bird, K. et al. [19].

**Figure 2 pathogens-14-00442-f002:**
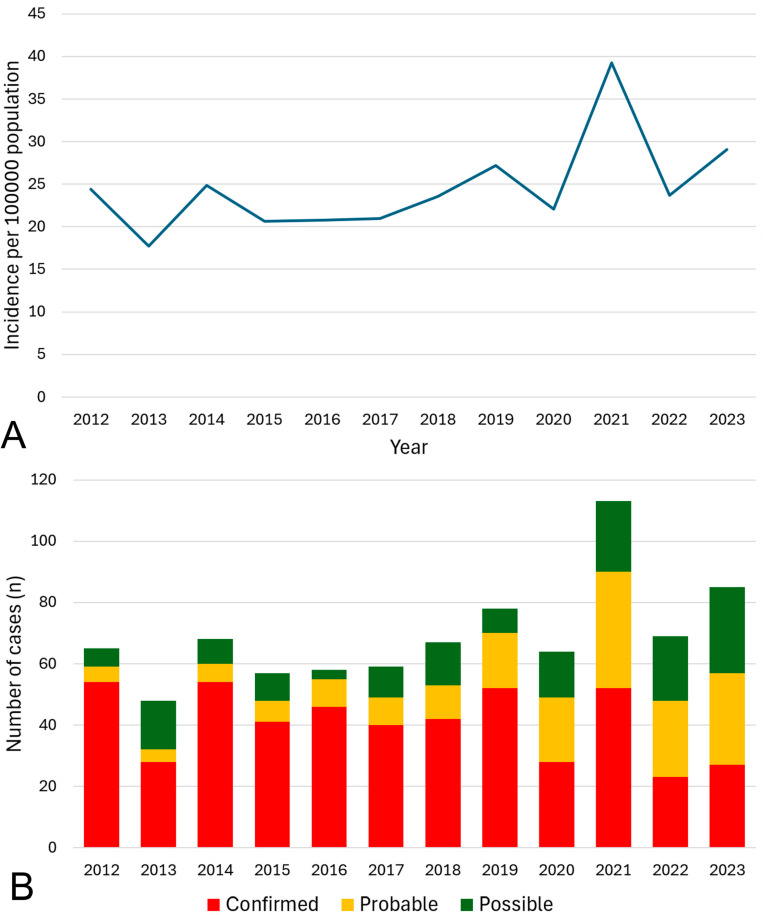
ARF cases in Far North Queensland diagnosed between January 2012 to December 2023. The incidence of all episodes of ARF in the general population is presented in panel (**A**). The proportion of episodes that were confirmed, probable and possible is presented in panel (**B**).

**Figure 3 pathogens-14-00442-f003:**
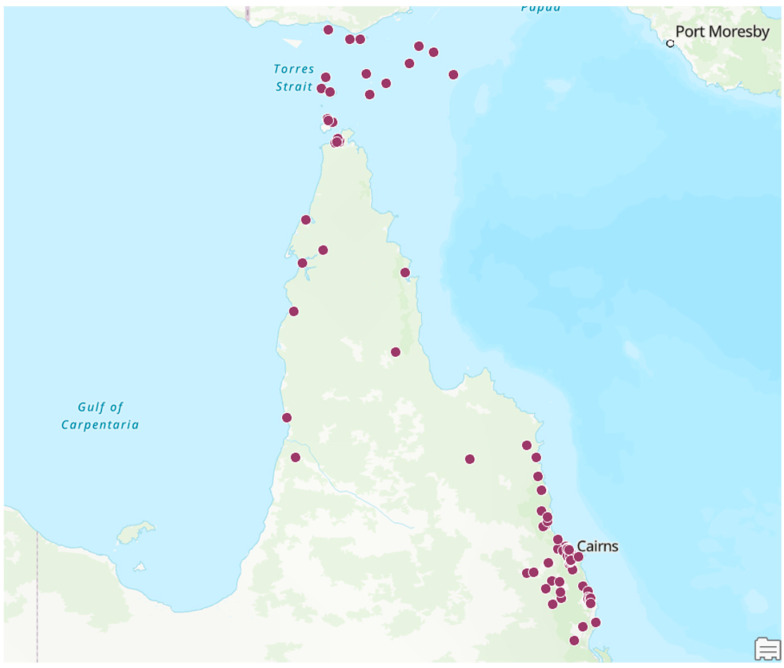
Geographical distribution of episodes of ARF (both initial and recurrent) in Far North Queensland from January 2012 to December 2023.

**Figure 4 pathogens-14-00442-f004:**
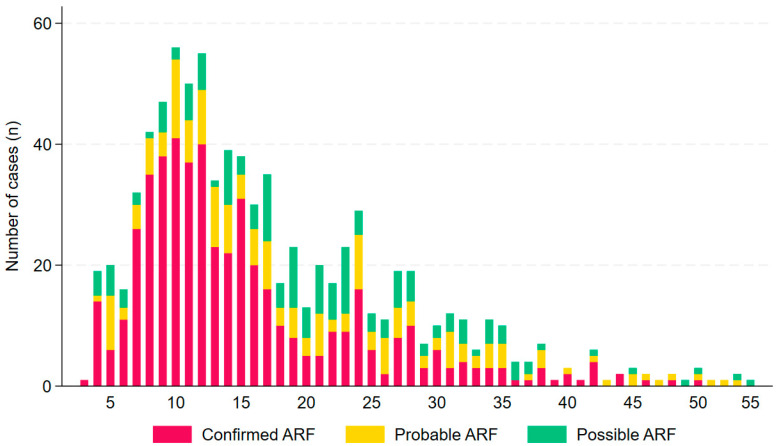
Age at ARF onset, by case definition in Far North Queensland January 2012 to December 2023.

**Figure 5 pathogens-14-00442-f005:**
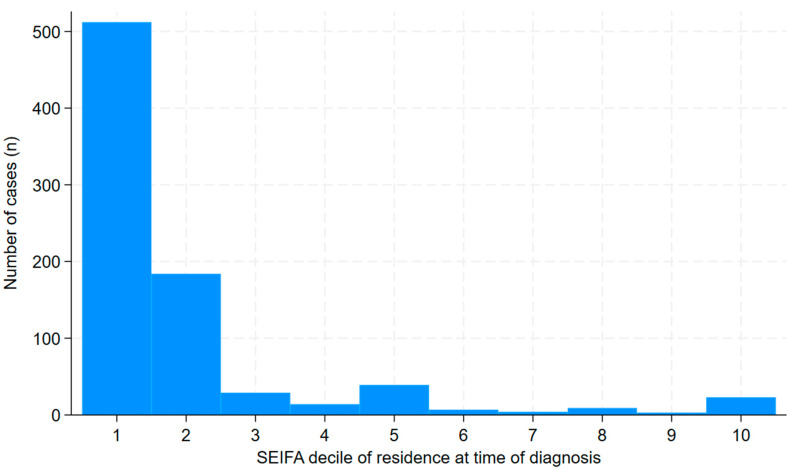
Socio-Economic Disadvantage of the episodes of ARF (both initial and recurrent) in Far North Queensland January 2012 to December 2023 (decile of ISRAD score presented) [20].

**Figure 6 pathogens-14-00442-f006:**
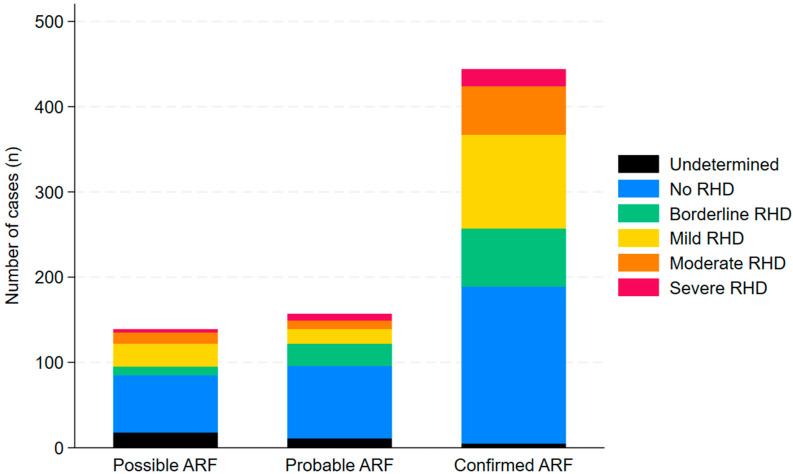
Prevalence and severity of RHD on the latest echocardiogram in individuals diagnosed with ARF in Far North Queensland, 2012 to 2023.

**Table 1 pathogens-14-00442-t001:** Major and minor manifestations for the diagnosis of acute rheumatic fever in Australia [15].

High-Risk Individuals ^a^	All Other Individuals
Major manifestations	
carditis (including subclinical echocardiography changes) polyarthritis, aseptic monoarthritis or polyarthralgia Sydenham chorea ^b^ erythema marginatum subcutaneous nodules	carditis (including subclinical echocardiography changes) polyarthritis Sydenham chorea ^b^ erythema marginatum subcutaneous nodules
Minor manifestations	
fever of 38 °C or higher monoarthralgia ESR 30 mm/hour or more, or CRP 30 mg/L or more prolonged PR interval on ECG	fever of 38.5 °C or higher polyarthralgia or aseptic monoarthritis ESR 60 mm/hour or more, or CRP 30 mg/L or more prolonged PR interval on ECG

^a^ High-risk individuals include: an Aboriginal and/or Torres Strait Islander person living in a rural or remote area; an Aboriginal and/or Torres Strait Islander person, or a Māori and/or Pacific person, living in a household affected by overcrowding or experiencing socioeconomic disadvantage; a person with a history of acute rheumatic fever or rheumatic heart disease; part of a family or household in which a member has a recent history of acute rheumatic fever or rheumatic heart disease; living in a household affected by overcrowding and experiencing socioeconomic disadvantage; current or prior residence in, or frequent or recent travel to, an Australian setting with a high rate of acute rheumatic fever; current or prior residence in, or frequent or recent travel to, an international setting with a high rate of acute rheumatic fever. ^b^ If Sydenham chorea is present, provided other causes of chorea are excluded, other clinical manifestations of acute rheumatic fever and evidence of preceding *S. pyogenes* infection are not required to make a diagnosis of definite acute rheumatic fever.

**Table 2 pathogens-14-00442-t002:** Demographic characteristics of episodes of ARF (both initial and recurrent) in Far North Queensland from January 2012 to December 2023.

	All n = 830	Confirmed n = 492	Probable n = 180	Possible n = 158
Age				
** <5 years**	20 (2%)	15 (3%)	1 (1%)	4 (3%)
** 5–9 years**	157 (19%)	116 (24%)	25 (14%)	16 (10%)
** 10–14 years**	234 (28%)	163 (33%)	47 (26%)	24 (15%)
** 15–19 years**	143 (17%)	85 (17%)	26 (14%)	32 (20%)
** 20–24 years**	102 (12%)	44 (9%)	24 (13%)	34 (22%)
** ≥25 years**	174 (21%)	69 (14%)	57 (32%)	48 (30%)
**Population group**				
** Aboriginal**	429 (52%)	248 (50%)	96 (53%)	85 (54%)
** Torres Strait Islander**	237 (29%)	145 (29%)	52 (29%)	40 (25%)
** Aboriginal and Torres Strait Islander**	119 (14%)	64 (13%)	25 (14%)	30 (19%)
** Māori**	6 (1%)	5 (1%)	1 (1%)	0
** Pacific Islander**	12 (1%)	11 (2%)	0	1 (1%)
** Other High Risk**	13 (1%)	10 (2%)	3 (2%)	0
** Other Low Risk**	14 (1%)	9 (2%)	3 (2%)	2 (1%)
**Sex**				
** Female**	475 (57%)	264 (54%)	111 (62%)	100 (63%)
** Male**	355 (43%)	228 (46%)	69 (38%)	58 (37%)
**Geography**				
** CHHHS**	371 (45%)	217 (44%)	88 (49%)	66 (42%)
** TCHHS**	459 (55%)	275 (56%)	92 (51%)	92 (58%)

CHHHS: Cairns and Hinterland Hospital and Health Service; TCHHS: Torres and Cape Hospital and Health Service.

**Table 3 pathogens-14-00442-t003:** Clinical and laboratory characteristics of ARF episodes (both initial and recurrent) and their association with a confirmed ARF diagnosis—as opposed to a non-confirmed diagnosis (probable or possible diagnosis)—in the cohort.

	All n = 830	Confirmed n = 492	Probable n = 180	Possible n = 158	Odds ratio ^a^ (95% CI)	*p*
**Major Criteria**						
Polyarthralgia	402 (48%)	231 (47%)	95 (53%)	76 (48%)	0.86 (0.66–1.14)	0.30
Monoarthritis	99 (12%)	59 (12%)	17 (9%)	23 (15%)	1.02 (0.66–1.56)	0.95
Polyarthritis	181 (22%)	130 (26%)	37 (21%)	14 (9%)	**2.02 (1.41–2.89)**	**<0.001**
Carditis	104 (13%)	89 (18%)	13 (7%)	2 (1%)	**4.76 (2.70–8.38)**	**<0.001**
Sydenham’s Chorea	53 (6%)	53 (11%)	0	0	-	-
Erythema Marginatum	13 (2%)	8 (2%)	4 (2%)	1 (1%)	1.10 (0.36–3.39)	0.87
Subcutaneous Nodules	10 (1%)	6 (1%)	3 (2%)	1 (1%)	1.03 (0.29–3.68)	0.96
**Minor Criteria**						
Monoarthralgia	67 (8%)	32 (7%)	14 (8%)	21 (13%)	**0.60 (0.36–0.99)**	**0.047**
Fever	409 (49%)	295 (60%)	69 (38%)	45 (28%)	**2.94 (2.20–3.93)**	**<0.001**
Prolonged PR interval	196 (24%)	161 (33%)	23 (13%)	12 (8%)	**4.21 (2.83–6.27)**	**<0.001**
CRP > 30 mg/L	510 (61%)	358 (73%)	104 (58%)	48 (30%)	**3.27 (2.44–4.38)**	**<0.001**
ESR > 30 mm/hr	450 (54%)	296 (60%)	102 (57%)	52 (33%)	**1.80 (1.36–2.39)**	**<0.001**
**Evidence of GAS infection**						
Median (IQR) ASOT (U/mL)	649 (385–1053)	768 (496–1190)	578 (347–1000)	395 (223–597)	**1.10 (1.07–1.14) ^b^**	**<0.001**
Median (IQR) Anti-DNase B (U/mL)	610 (389–957)	650 (429–1015)	586 (389–954)	391 (257–735)	**1.06 (1.03–1.10) ^b^**	**<0.001**
Positive skin swab	43 (5%)	26 (5%)	9 (5%)	8 (5%)	1.05 (0.56–1.97)	0.87
Positive throat swab	70 (8%)	43 (9%)	15 (8%)	12 (8%)	1.10 (0.67–1.82)	0.70
Positive blood culture	1 (0.1%)	1 (0.2%)	0	0	-	-
Stated history	166 (20%)	96 (20%)	43 (24%)	27 (17%)	0.93 (0.66–1.31)	0.67
Unknown	550 (66%)	326 (66%)	113 (63%)	111 (70%)	1.00 (0.75–1.34)	1.0

^a^ Odds ratio for the presence of this finding in cases of confirmed ARF compared with non-confirmed ARF (probable or possible diagnosis) ^b^ For every increase of 100 U/mL

## Data Availability

Data cannot be shared publicly because of the Queensland Public Health Act 2005. Data are available from the Far North Queensland Human Research Ethics Committee (contact via email FNQ_HREC@health.qld.gov.au) for researchers who meet the criteria for access to confidential data.

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
