# Peer review of "Epidemiological and Clinical Manifestations of Acute Rheumatic Fever in Far North Queensland, Australia"

_pathogens, 2025, doi:10.3390/pathogens14050442_

Round 1

Reviewer 1 Report

Comments and Suggestions for Authors

Epidemiological and clinical characteristics of individuals with acute rheumatic fever in Far North Queensland, Australia.

I found the manuscript interesting, and you can find my report, with appraisal, suggestions, and concerns, section by section as follows:

The abstract is well-written, and it is informative. I suppose that rs means rho of Spearman, but I do not know if it is possible to use rs as rho. However, the abstract needs to be modified after the revision process.

Introduction: The ARF is well-explained from a clinical and epidemiological point of view.

Interestingly, you also highlighted the ethnic differences in a country, Australia, which has several ethnic groups and immigration from different parts of the world (Europe, Asia, Africa, North and South America). Despite this, since you have taken into consideration only data from 2012 to 2023, you should introduce the reason for this choice. It is not clear if previous research is present or if data were lacking. Please, add a brief statement. Moreover, the aims at the end of the introduction are concise, but I should avoid the “hope sentence” or I invite you to rewrite it in a better way.  

The methods are well-written and allow the replication with data from different countries. However, in the setting, more information about the population in demographic terms is suggested, since FAAs seems to be the focus. Moreover, a discrepancy appears between the whole population in the area and the FAA (17%), and 12%  and 69% (of 17%) are served by two different health services. This is not completely clear when you read the text. Please, rewrite it.  I agree with the statistical analyses, I advise giving more information about logistic regression and about the models used. The results are shown in a good way, with informative tables and figures.

Discussion: I agree with the summary of the results at the start of the section, but I’d avoid the comparison with the UK since the two countries have different climatic and socioeconomic situations. Your results were discussed in a good way in light of previously published literature. Future directions and limitations were also discussed in a good way, as well as the conclusion. 

Author Response

Reviewer 1

I found the manuscript interesting, and you can find my report, with appraisal, suggestions, and concerns, section by section as follows:

Response: We thank Reviewer 1 for the time that he/she has taken to review our manuscript and the helpful suggestions that he/she has made for its enhancement. Please find our point-by-point response to his/her comments below.

The abstract is well-written, and it is informative. I suppose that rs means rho of Spearman, but I do not know if it is possible to use rs as rho. However, the abstract needs to be modified after the revision process.

Response: The Reviewer is absolutely correct; we did indeed use rs as an abbreviation for Spearman’s rho. We have revised the abstract to use the full term “Spearman’s rho” to avoid any confusion.

Change: Revision of abstract; replacement of “rs with “Spearman’s rho”

Introduction: The ARF is well-explained from a clinical and epidemiological point of view.

Response: We are grateful to hear that the Reviewer appears happy with the introduction.

Interestingly, you also highlighted the ethnic differences in a country, Australia, which has several ethnic groups and immigration from different parts of the world (Europe, Asia, Africa, North and South America). Despite this, since you have taken into consideration only data from 2012 to 2023, you should introduce the reason for this choice. It is not clear if previous research is present or if data were lacking. Please, add a brief statement. Moreover, the aims at the end of the introduction are concise, but I should avoid the “hope sentence” or I invite you to rewrite it in a better way.  

Response: We thank the Reviewer for raising this issue. Again, he/she is correct to suggest that we chose the time period of 2012-2023 due to incomplete data prior to 2012. We have added a sentence to the methods to clarify this.

We agree that the verb “hope” is poorly chosen here. We have revised the text as the Reviewer has suggested, to emphasise that, specifically, these data will be disseminated to inform the earlier recognition and the optimal management of cases of ARF in the FNQ region.

Change: Revision of methods to explain why we have chosen the 2012-23 time period. Revision of the introduction, replacing the phrase containing “hope”.

The methods are well-written and allow the replication with data from different countries. However, in the setting, more information about the population in demographic terms is suggested, since FAAs seems to be the focus. Moreover, a discrepancy appears between the whole population in the area and the FAA (17%), and 12%  and 69% (of 17%) are served by two different health services. This is not completely clear when you read the text. Please, rewrite it.  I agree with the statistical analyses, I advise giving more information about logistic regression and about the models used. The results are shown in a good way, with informative tables and figures.

Response: We thank the Reviewer for raising this issue. The reason that we focus on First Nations Australians in the paper is, as we point out in the introduction: “The burden of ARF is borne almost entirely by its Aboriginal and Torres Strait Islander peoples (hereafter respectfully referred to as First Nations peoples). While the overall incidence of ARF in Australia is 69 per 100,000, the incidence in First Nations peoples is up to 121 times that of other Australians.

This is emphasised in our data where is that 785/830 cases of ARF occurred in First Nations Australians (as we highlight in the population group section in table 2) and the text in the discussion (lines XXX). However, we have expanded the methods to provide more of a “snapshot” of the FNQ population (using 2021 Australian census data).

We feel that if the Reviewer reads the methods carefully, he/she will note that 12% of ~265000 (~32000) + 69% of ~25000 (17000) = 49000 which is ~17% of the region’s total population of 290,000. However, we recognise that we are more familiar with the regions that the reader and so we have added the numbers to the text to make the calculation of the different populations absolutely transparent.

Change: Revision of methods to describe the demography ion more detail. Inclusion of the absolute numbers of First Nations Australians in each health service to help the reader understand (precisely) the distribution of First Nations Australians in the FNQ region.

Discussion: I agree with the summary of the results at the start of the section, but I’d avoid the comparison with the UK since the two countries have different climatic and socioeconomic situations. Your results were discussed in a good way in light of previously published literature. Future directions and limitations were also discussed in a good way, as well as the conclusion. 

Response: We thank the Reviewer for raising this issue. We were hoping to convey to the reader the sheer size of the FNQ region and the challenges with delivering optimal care across such a vast area. However, clearly this has caused some confusion and so we have revised the text to delete all reference to the United Kingdom, instead highlighting the actual size (in square kilometres) of the FNQ region.

We are pleased to hear that the Reviewer was happy with the future directions, limitations and conclusions.

Change: Revision of the discussion. Deletion of all reference to the United Kingdom.

Reviewer 2 Report

Comments and Suggestions for Authors

The authors present an epidemiologic and descriptive study of ARF and RHD characteristic in a population from a specific geographic area. Here you are my comments:

  • The reason why the paper is of certain interest, what is the added info to the literature and therefore why it should be published, should be clearly expressed in the discussion at least.
  • Table 4 presents OR: however the reference group should always be specified when reporting OR. 

Author Response

Reviewer 2

The authors present an epidemiologic and descriptive study of ARF and RHD characteristic in a population from a specific geographic area. Here you are my comments:

Response: We thank Reviewer 2 for the time that he/she has taken to review our manuscript and the helpful suggestions that he/she has made for its enhancement. Please find our point-by-point response to his/her comments below.

The reason why the paper is of certain interest, what is the added info to the literature and therefore why it should be published, should be clearly expressed in the discussion at least.

Response: We thank the Reviewer for raising this issue. The main focus of our paper was to examine epidemiological and clinical data from the Queensland ARF and RHD register to characterise ARF presentations in the FNQ population in order to facilitate the prompt local diagnosis of ARF and the more timely delivery of the multi-faceted care that is required to reduce the local incidence of RHD and its complications. We feel that we have done this.

However, in the revised manuscript we have expanded the discussion to highlight the specific issues that we feel our paper highlights (which are also presented in separate paragraphs in the discussion) which add to the literature.

Specifically, we have provided contemporary data that highlights:

  • The clinical and laboratory findings that are most associated with a diagnosis of confirmed ARF.
  • Common patterns of joint involvement in individuals with ARF in Australia.
  • We highlight that while children aged 5-14 are the population that are most likely to have confirmed disease, ARF is not uncommon in older populations.
  • The manuscript also provides additional (albeit circumstantial) data to suggest that there is a role for GAS infection of the skin in the pathogenesis of ARF.
  • The manuscript also describes, quantitatively, the association between ARF and socioeconomic disadvantage in the region (using Australian Bureau of Statistics SEIFA score).

Change: Expansion of the discussion to highlight, specifically, the ways in which our paper adds to the literature.

Table 4 presents OR: however the reference group should always be specified when reporting OR. 

Response: We thank the Reviewer for raising this issue (although we think they mean table 3 as we do not have  a table 4?)

If the Reviewer looks carefully, he/she will note that footnote “a” highlights that the reference group (dependent variable) is those with confirmed ARF, However, in the revised manuscript we have amended the title of the table and expanded the footnote to make this point more clearly.

Change: Revision of the title and footnote of Table 3 to highlight that confirmed ARF was the dependent variable.

Round 2

Reviewer 1 Report

Comments and Suggestions for Authors

You have satisfactorily addressed my concerns.

Author Response

Reviewer 1.

You have satisfactorily addressed my concerns.

Response: We thank Reviewer 1 for the time that he/she has taken to review our paper and the helpful advice for its enhancement. 

Reviewer 2 Report

Comments and Suggestions for Authors

Thanks to the authors for revising the manuscript. However, my queries have not been addressed properly. For instance:

Authors' response: In the revised manuscript we have expanded the discussion to highlight the specific issues that we feel our paper highlights (which are also presented in separate paragraphs in the discussion) which add to the literature.

Reviewer's response: if you do not provide any new information to the literature (besides the well known fact that low-income people have a higher risk of dealing with a disease that is based on an infection). I highly recommend to be crystal clear about this aspect. Otherwise your study represents nothing new and it is hard to accept. 

Authors' response: If the Reviewer looks carefully, he/she will note that footnote “a” highlights that the reference group (dependent variable) is those with confirmed ARF.

Reviewer's response: the reference group is the one on the denominator. If you consider the confirmed ARF as the reference group, then, in example, the 2.02 OR for polyarthritis or 4.76 for carditis in table 3 (sorry for my mistake) is in favour of possible or probable ARF whereas an OR can only be calculated comparing two groups (i.e. confirmed vs possible as the reference group). Please revise your statement and be clear. 

Added comments:

  • The statistical analysis section should be expanded with more info about the different groups you decided to compare, the variables' (normal vs not) distribution and tests you used.
  • I recommend to split the 2 figure 2 in a and b.
  • I suggest to add to table 3 a column with total population numbers. 

Author Response

We thank Reviewer 2 for the time that he/she has taken to review our revised manuscript and the helpful suggestions for its enhancement.

Please find our point-by-point reply to his/her comments (responses follow the bolded "Response to round 2 of Reviewer 2’s comments:")

Reviewer 2’s comments

Thanks to the authors for revising the manuscript. However, my queries have not been addressed properly. For instance:

Authors' response: In the revised manuscript we have expanded the discussion to highlight the specific issues that we feel our paper highlights (which are also presented in separate paragraphs in the discussion) which add to the literature.

Reviewer's response: if you do not provide any new information to the literature (besides the well known fact that low-income people have a higher risk of dealing with a disease that is based on an infection). I highly recommend to be crystal clear about this aspect. Otherwise your study represents nothing new and it is hard to accept. 

Response to round 2 of Reviewer 2’s comments: We are sorry that the Reviewer did not feel that we addressed his/her queries properly.

In our defence, we did not only highlight that socio-economically deprived populations have a higher incidence of ARF (although we feel that this is an issue that is essential to highlight in 21st century Australia), there was also a significant focus on the clinical presentation of ARF in this unique region of Australia and the features that were most associated with confirmed disease.

The study is the largest to examine the clinical presentation of ARF in the FNQ region of Australia which is unique as it is the only region of Australia which has the homelands of both Aboriginal and Torres Strait Islander Australians.

The points that we highlight in the paper are the importance of seeking carditis (which had a greater association with confirmed ARF), the patterns of arthritis associated with confirmed disease (polyarthritis but not monoarthritis, polyarthralgia or mono-arthralgia), the utility of laboratory variables in establishing a confirmed diagnosis of ARF, and the incidence of ARF in individuals outside the classic age 5-14 age group (43% of confirmed ARF occurred outside this age group). We also highlight that a significant proportion of individuals with preceding microbiological evidence of GAS infection had skin infection as opposed to pharyngitis (5% versus 8%). This could certainly be described as “new” information as there is still controversy about the role of GAS infection of the skin in the pathogenesis of ARF/RHD.

We also emphasise that the incidence of confirmed ARF declined over the study period, which is at least indirect evidence that the ongoing diagnosis of ARF is due to enhanced awareness of the condition, suggesting that efforts to educate local clinicians about ARF are having an impact.

However, clearly the Reviewer feels that we did not make these points clearly enough in the prior submission. We have therefore revised the discussion extensively to highlight the novel aspects of our work (recognising that, of course, very little scientific literature is completely novel!).

We don’t feel that it possible (or appropriate) to talk about ARF in Australia without highlighting the social determinants of health and the disproportionate burden of disease that is experienced by its First Nations people. However, we have relegated this discussion to the end of the manuscript to address the Reviewer’s concerns.

Change: Extensive revision of the discussion highlighting the novel findings in our cohort in this unique region of Australia. Specifically, the Jones criteria associated with a confirmed ARF diagnosis, the high proportion of older adults with a confirmed ARF diagnosis and the role of GAS skin infection in the pathogenesis of ARF in the region.

Authors' response to round 1: If the Reviewer looks carefully, he/she will note that footnote “a” highlights that the reference group (dependent variable) is those with confirmed ARF.

Reviewer's response: the reference group is the one on the denominator. If you consider the confirmed ARF as the reference group, then, in example, the 2.02 OR for polyarthritis or 4.76 for carditis in table 3 (sorry for my mistake) is in favour of possible or probable ARF whereas an OR can only be calculated comparing two groups (i.e. confirmed vs possible as the reference group). Please revise your statement and be clear. 

Response to round 2 of Reviewer 2’s comments: We thank the Reviewer for raising this issue and apologise that we were not clearer. The two groups that we compared were confirmed and non-confirmed (probable and possible combined). We have revised the figure legend and footnote to make this clearer.  

Change: Revision of figure legend and footnote to highlight that the two groups were confirmed and non-confirmed (possible and probable).

Added comments:

  • The statistical analysis section should be expanded with more info about the different groups you decided to compare, the variables' (normal vs not) distribution and tests you used.

Response to round 2 of Reviewer 2’s comments: We thank the reviewer for this suggestion. We have added text to the statistical section as suggested, highlighting that the distribution of age and the serological titres were non-parametric. We had described the two statistical tests that we used in the paper in the original draft (logistic regression to calculate odds ratios and Spearman’s test for trends over time).

  • I recommend to split the 2 figure 2 in a and b.

Response to round 2 of Reviewer 2’s comments: We thank the reviewer for this suggestion. We have revised figure 2 as suggested.

  • I suggest to add to table 3 a cumn with total population numbers. 

Response to round 2 of Reviewer 2’s comments: We thank the reviewer for this suggestion. We have revised table 3, adding a column with total population numbers as suggested.

Round 3

Reviewer 2 Report

Comments and Suggestions for Authors

Article significantly improved, 

suitable for publication. 

Congratulation.